**Data Availability Statement:** All relevant data are within the manuscript and its Supporting Information files.

# Evaluating the diagnostic accuracy of the WHO Severe Acute Respiratory Infection (SARI) criteria in Middle Eastern children under two years over three respiratory seasons

Thomas Klink[1], Danielle A. Rankin[2,3], Bhinnata Piya[4], Andrew J. Spieker[5], Samir Faouri[6], Asem Shehabi[7], John V. Williams[8], Najwa Khuri-Bulos[7‡], Natasha B. Halasa[3‡*]

1 Medicine and Pediatric Residency Program, University of Minnesota, Minneapolis, Minnesota, United States of America, 2 Vanderbilt Epidemiology PhD Program, Vanderbilt University School of Medicine, Vanderbilt University, Nashville, Tennessee, United States of America, 3 Department of Pediatrics, Vanderbilt University, Nashville, Tennessee, United States of America, 4 Vanderbilt Institute for Clinical and Translational Research, Nashville, Tennessee, United States of America, 5 Department of Biostatistics, Vanderbilt University, Nashville, Tennessee, United States of America, 6 Department of Pediatrics, Al Bashir Hospital, Amman, Jordan, 7 Department of Pediatrics, Jordan University, Amman, Jordan, 8 Department of Pediatrics, University of Pittsburgh School of Medicine, Children's Hospital of Pittsburgh of University of Pittsburgh Medical Center, Pittsburgh, Pennsylvania, United States of America

‡ These authors are joint senior authors on this work.
* Natasha.halasa@vumc.org

## Abstract

### Objective

The World Health Organization created the Severe Acute Respiratory Infection (SARI) criteria in 2011 to monitor influenza (flu)-related hospitalization. Many studies have since used the SARI case definition as inclusion criteria for surveillance studies. We sought to determine the sensitivity, specificity, positive predictive value, and negative predictive value of the SARI criteria for detecting ten different respiratory viruses in a Middle Eastern pediatric cohort.

### Materials and methods

The data for this study comes from a prospective acute respiratory surveillance study of hospitalized children <2 years in Amman, Jordan from March 16, 2010 to March 31, 2013. Participants were recruited if they had a fever and/or respiratory symptoms. Nasal and throat swabs were obtained and tested by real-time RT-PCR for eleven viruses. Subjects meeting SARI criteria were determined post-hoc. Sensitivity, specificity, positive predictive value, and negative predictive value of the SARI case definition for detecting ten different viruses were calculated and results were stratified by age.

### Results

Of the 3,175 patients enrolled, 3,164 were eligible for this study, with a median age of 3.5 months, 60.4% male, and 82% virus-positive (44% RSV and 3.8% flu). The sensitivity and

**Funding:** N.H. received grants from Sanofi and Quidel and consulting fees for Moderna, and Karius outside the submitted work. J.V.W. reports personal fees from Quidel, personal fees from GlaxoSmithKline outside the submitted work. For the remaining authors, none were declared. This work was supported by the UBS Optimus Foundation; National Institutes of Health: R01AI085062, and the CTSA award UL1TR000445 from the National Center for Advancing Translational Sciences. Its contents are solely the responsibility of the authors and do not necessarily represent official views of the National Center for Advancing Translational Sciences or the National Institutes of Health.

**Competing interests:** N.H. received grants from Sanofi and Quidel and consulting fees for Moderna, and Karius outside the submitted work. J.V.W. reports personal fees from Quidel, personal fees from GlaxoSmithKline outside the submitted work. For the remaining authors, none were declared. This work was supported by the UBS Optimus Foundation; National Institutes of Health: R01AI085062, and the CTSA award UL1TR000445 from the National Center for Advancing Translational Sciences. Its contents are solely the responsibility of the authors and do not necessarily represent official views of the National Center for Advancing Translational Sciences or the National Institutes of Health.

specificity of the SARI criteria for detecting virus-positive patients were 44% and 77.9%, respectively. Sensitivity of SARI criteria for any virus was lowest in children <3 months at 22.4%. Removing fever as a criterion improved the sensitivity by 65.3% for detecting RSV in children <3 months; whereas when cough was removed, the sensitivity improved by 45.5% for detecting flu in same age group.

## Conclusions

The SARI criteria have poor sensitivity for detecting RSV, flu, and other respiratory viruses —particularly in children <3 months. Researchers and policy makers should use caution if using the criteria to estimate burden of disease in children.

## Introduction

Respiratory infections are the second leading cause of global years of life lost in all ages, and the leading cause of mortality in children under five years [1]. In 2011, the World Health Organization (WHO) created a case definition for Severe Acute Respiratory Infection (SARI) in an attempt to standardize global surveillance of hospitalization related to influenza (flu)—allowing national health authorities to interpret their data in an international context [2]. Flu is particularly difficult to surveil because its clinical presentation is often indistinguishable from other respiratory viruses [3]. With that in mind, the WHO designed the SARI criteria to strike a balance between sensitivity and specificity, while also noting that the case definition is not necessarily intended to capture all cases but to describe trends over time [4].

Several studies, including at least nine in the Eastern Mediterranean region, have been published since 2011 using the SARI case definition as inclusion criteria to report a combination of clinical characteristics, risk factors, viral burden, or outcomes in adult and pediatric populations for flu and other respiratory viruses [5–13]. Only a handful of studies have evaluated the effectiveness of the criteria by including both SARI-positive and SARI-negative patients, allowing them to calculate the diagnostic accuracy of the criteria for detecting flu and respiratory syncytial virus (RSV) [14–19]. The diagnostic accuracy of the criteria for other individual respiratory viruses remains unknown. Additionally, only one of these studies stratified age to include a group of children less than three months old, and studies of this type from the Eastern Mediterranean region are lacking [17].

Moreover, those studies that have included both SARI-positive and SARI-negative pediatric patients have indicated that the SARI criteria are less sensitive in younger pediatric patients, but only one such study stratified by the youngest children [14, 16–18]. This means that studies using the SARI case definition as inclusion criteria may underestimate the burden of disease in this age group [5]. Therefore, we sought to evaluate the diagnostic accuracy of the SARI criteria for eleven respiratory viruses in a large hospitalized pediatric cohort in Amman, Jordan in children less than two years old who presented with fever and/or respiratory symptoms.

## Materials and methods

### Study design and participants

We conducted a prospective active surveillance study of acute respiratory infections in hospitalized children <2 years in Amman, Jordan. Participants were recruited over a three-year period (March 16, 2010-March 31, 2013) within 48 hours of hospital admission for fever and/

or respiratory symptoms with one of the following admission diagnoses: ARI, apnea, asthma exacerbation, bronchiolitis, bronchopneumonia, croup, cystic fibrosis exacerbation, febrile seizure, fever without localizing signs, respiratory distress, pneumonia, pneumonitis, pertussis, pertussis-like cough, rule out sepsis, upper respiratory infection (URI), or other. Children were excluded only if they had chemotherapy-associated neutropenia and/or were newborns who had never been discharged [20]. Trained local staff obtained written informed consent from parents or guardians of all participants. The study was approved by the Institutional Review Boards of the University of Jordan, the Jordanian Ministry of Health, and Vanderbilt University.

## Setting

Al-Bashir Hospital is a 185 pediatric bed (120 pediatric and 65 neonatal intensive care unit) government-run hospital that serves the Jordanian capital, Amman, a city with >2 million inhabitants. The catchment area is densely populated, low-income, and includes a Palestinian refugee camp. Due to government policy, children <6 years are provided free care at Al-Bashir regardless of insurance status. During the study period, there were 11,230 hospitalizations of children <2 years.

## Data and specimen collection

Throat and nasal swabs from all participants were obtained by trained research staff. Demographic, social, and medical histories were obtained by standardized questionnaires, which were a component of the interview portion of the case report form. All interviews were conducted in Arabic to parents and recorded in English. Following patient discharge, clinical outcome data and antibiotic use during the hospitalization data were systematically collected from the medical record. All data were entered into a secure REDCap[TM] (Research Electronic Data Capture, Vanderbilt University, Nashville, TN, USA) database. Extracted information is explained in detail in a previous publication [20].

## Laboratory testing

Throat and nasal swabs were combined in transport medium (M4RT[®], Remel, USA) aliquoted into MagMax[TM] Lysis/Binding Solution Concentrate (Life Technologies, USA), snap frozen, stored at -80°C, and shipped to Nashville, TN. Original and lysis buffers were tested by real-time RT-PCR for eleven respiratory viruses: RSV, human metapneumovirus (HMPV), human rhinovirus (HRV), flu A, B, and C, parainfluenza virus (PIV) 1, 2, and 3, adenovirus, and Middle East respiratory syndrome coronavirus (MERS-CoV) [21]. Cycle threshold (Ct) values were determined, with lower Ct corresponding with higher viral load.

## SARI case definition

The WHO SARI case definition includes three elements: hospitalization with ARI with symptom onset in the past 10 days, history of fever or measured fever $\geq$38 C°, and cough [4]. In order to capture this definition in our subjects, we used information obtained from the questionnaire and extracted from the medical chart. Days of symptoms were captured in the questionnaire. Fever was captured with at least one of the following: an admission or discharge diagnosis of fever, measured temperature $\geq$38 C° at admission, or reported history of fever as a symptom of current illness. Cough was recorded if it was reported as a symptom or was captured as an admission or discharge diagnosis.

## Statistical analysis

Descriptive statistics were reported as frequencies or mean, median and interquartile range (IQR) where appropriate. Odds ratios for covariates contributing to SARI status were calculated using a series of simple logistic regressions, followed by multiple logistic regressions, controlling for age and sex. Holm-Bonferroni adjustments were made to account for multiple analyses with the same dependent variable. Covariates included age, sex, birthweight, prematurity, medical history, smoke exposure [nargila (smoking pipe) or cigarettes], vitamin D level, breastfeeding, length of illness, admission diagnoses of pneumonia, bronchopneumonia, bronchiolitis, sepsis, or febrile seizure, viral detection, and markers of illness severity including antibiotics before or after hospitalization, length of stay, ICU admission, oxygen use, mechanical ventilation, and death. All analyses were completed using STATA version 15.1.

Sensitivity, specificity, positive predictive value (PPV), and negative predictive value (NPV) of the SARI criteria and modifications excluding either fever or cough were calculated for each respiratory virus tested. Parainfluenza viruses 1, 2, and 3 were combined into one category, as was flu A, B, and C. This was done for simplicity and relatively low numbers of detections of each subtype. For NPV and PPV, we assumed the prevalence found in our study to be accurate for hospitalized children with fever or respiratory symptoms. For each virus, the prevalence of cough and fever was also calculated. These data were then stratified by age into three groups: less than 3 months, 3–5 months, and 6–23 months. These groups were chosen to compare results to a previous study; and were based off a visual analysis of the age of virus-positive patients meeting SARI criteria [17].

# Results

## Study population

Between March 16, 2010, and March 31, 2013, there were 3,793 eligible hospitalized infants and 3,175 (83.7%) were enrolled as previously described [20]. Seven of these were excluded: four admitted with the diagnosis of meningitis and three who were older than two years. Of the remaining 3,168 subjects, four were excluded for the purpose of this analysis because they did not have information on the SARI criterion of illness duration. Therefore, 3,164 subjects (83.4% of eligible patients) are included in our analyses.

The median age for enrolled subjects was 3.5 months (**Table 1**). Most were male (60.4%), had household exposure to smoke or nargila (76.6%), and reported exclusive breastfeeding (60.6%). Viral pathogens were detected in 2,581 (81.5%) subjects, and 315 (10.1%) had at least one underlying medical condition.

## Clinical and demographic characteristics of SARI-positive vs. SARI-negative patients after adjustment for age and sex

Of the 3,164 subjects included, 1,261 (39.9%) met SARI criteria (**Table 1**). When compared to their SARI-negative counterparts, SARI-positive subjects tended to be older, were less likely to have a history of premature birth, and had a shorter duration of illness prior to hospitalization. No significant differences were detected in sex, underlying medical conditions, smoke exposure, vitamin D levels, birth weight, or breastfeeding history.

SARI-positive patients were more likely to be diagnosed with pneumonia (OR: 1.77; 95% CI: 1.41–2.22; p-value: <0.001) and bronchopneumonia (OR: 3.83; 95% CI: 3.21–4.55; p-value: <0.001). Alternatively, SARI-negative subjects were more likely to be diagnosed with rule out sepsis (OR: 0.37; 95% CI: 0.20–0.47; p-value: <0.001) and febrile seizure (OR: 0.07; 95% CI: 0.03–0.14; p-value: <0.001). No significance was found in the prevalence of bronchiolitis

**Table 1. Demographics and clinical characteristics by SARI status.**

| | All (n = 3164) | SARI-Positive (n = 1261) | SARI-Negative (n = 1903) | Unadjusted Odds Ratio (95% CI) | p-value | Adjusted Odds Ratio* (95% CI) | p-value |
|---|---|---|---|---|---|---|---|
| Age (months, median) | 3.49 (1.64–8.48) 5.78 | 6.48 (3.4–11.6) 7.95 | 2.30 (1.25–5.44) 4.34 | 1.13 (1.12–1.15) | <0.001[t] | 1.13 (1.12–1.15) | <0.001[t] |
| Sex (male) | 1910 (60.4) | 772 (61.2) | 1138 (59.8) | 1.06 (0.92–1.23) | 0.424 | 1.08 (0.93–1.26) | 0.307 |
| Premature, <37 weeks | 447 (14.1) | 144 (11.4) | 303 (16.0) | 0.68 (0.55–0.84) | <0.001[t] | 0.68 (0.54–0.85) | 0.001[t] |
| Birth Weight (kg, median) (n = 3162) | 3.0 (2.5–3.5) 2.97 | 3.0 (2.6–3.5) 3.0 | 3.0 (2.5–3.5) 3.0 | 1.12 (1.00–1.25) | 0.044 | 1.14 (1.01–1.28) | 0.027 |
| UMC** | 320 (10.1) | 141 (11.2) | 179 (9.41) | 1.21 (0.96–1.53) | 0.105 | 0.87 (0.68–1.12) | 0.269 |
| Smoke Exposure (Nargila or cigarette) | 2422 (76.6) | 970 (76.9) | 1452 (76.3) | 1.03 (0.87–1.22) | 0.686 | 1.04 (0.87–1.25) | 0.633 |
| Vitamin D Level, median | 16.5 (5.2–26) 17.1 | 19.4 (7.1–27.5) 18.8 | 14.5 (4.7–24.7) 16.0 | 1.02 (1.01–1.02) | <0.001[t] | 1.01 (1.00–1.01) | 0.024 |
| Exclusively Breastfed | 1918 (60.6) | 803 (63.7) | 1115 (58.6) | 1.24 (1.07–1.44) | 0.004 | 1.06 (0.91–1.24) | 0.478 |
| No. Days Sick, median | 3 (2–5) 4.03 | 3 (2–5) 3.74 | 3 (1–4) 4.22 | 0.99 (0.97–1.01) | 0.196 | 0.97 (0.95–0.99) | 0.001[t] |
| **Admission Diagnosis** | | | | | | | |
| Pneumonia | 394 (12.5) | 200 (15.9) | 194 (10.2) | 1.66 (1.34–2.05) | <0.001[t] | 1.77 (1.41–2.22) | <0.001[t] |
| Bronchopneumonia | 1018 (32.2) | 679 (53.9) | 339 (17.8) | 5.38 (4.58–6.33) | <0.001[t] | 3.83 (3.21–4.55) | <0.001[t] |
| Bronchiolitis | 546 (17.3) | 191 (15.2) | 355 (18.7) | 0.77 (0.64–0.94) | 0.011 | 0.96 (0.79–1.17) | 0.687 |
| Sepsis | 899 (28.4) | 157 (12.5) | 742 (39.0) | 0.22 (0.18–0.27) | <0.001[t] | 0.37 (0.30–0.46) | <0.001[t] |
| Febrile Seizure | 83 (2.62) | 10 (0.79) | 73 (3.8) | 0.20 (0.10–0.39) | <0.001[t] | 0.07 (0.03–0.14) | <0.001[t] |
| **Virus(es) Detected including co-detections** | | | | | | | |
| Any Virus | 2579 (81.5) | 1132 (89.8) | 1447 (76.0) | 2.77 (2.24–3.41) | <0.001[t] | 2.84 (2.27–3.56) | <0.001[t] |
| RSV | 1396 (44.1) | 675 (53.5) | 721 (37.9) | 1.89 (1.63–2.18) | <0.001[t] | 2.34 (2.00–2.74) | <0.001[t] |
| Rhinovirus | 1237 (39.1) | 457 (36.2) | 780 (41.0) | 0.82 (0.71–0.95) | 0.007 | 0.82 (0.70–0.96) | 0.013 |
| Adenovirus | 474 (15.0) | 207 (16.4) | 267 (14.0) | 1.20 (0.99–1.47) | 0.066 | 0.96 (0.78–1.19) | 0.711 |
| hMPV | 273 (8.63) | 160 (12.7) | 113 (6.0) | 2.30 (1.79–2.96) | <0.001[t] | 2.09 (1.60–2.72) | <0.001[t] |
| PIV 1–3 | 175 (5.53) | 77 (6.1) | 98 (5.2) | 1.19 (0.88–1.63) | 0.250 | 1.08 (0.78–1.50) | 0.648 |
| Flu A-C | 119 (3.76) | 63 (5.0) | 56 (2.9) | 1.73 (1.20–2.50) | 0.003 | 1.47 (0.99–2.18) | 0.053 |
| Co-Infection | 943 (29.8) | 433 (34.3) | 510 (26.8) | 1.43 (1.22–1.67) | <0.001[t] | 1.45 (1.23–1.70) | <0.001[t] |
| **Illness Severity** | | | | | | | |
| Abx before hospitalization | 1284 (40.6) | 641 (50.8) | 643 (33.8) | 2.03 (1.75–2.34) | <0.001[t] | 1.46 (1.25–1.72) | <0.001[t] |
| Abx during hospitalization | 2882 (91.1) | 1191 (94.4) | 1691 (88.9) | 2.13 (1.61–2.82) | <0.001[t] | 2.33 (1.73–3.12) | <0.001[t] |
| Length of Stay | 5 (3–7) 5.59 | 4 (3–7) 5.17 | 5 (3–8) 5.87 | 0.95 (0.94–0.97) | <0.001[t] | 0.98 (0.966–1.00) | 0.124 |
| ICU | 246 (7.85) | 72 (5.8) | 174 (9.2) | 0.60 (0.45–0.80) | <0.001[t] | 0.72 (0.54–0.97) | 0.033 |
| Oxygen | 1013 (32.3) | 403 (32.2) | 610 (32.4) | 0.98 (0.84–1.15) | 0.840 | 1.17 (1.00–1.38) | 0.057 |
| Mechanical Vent | 111 (3.54) | 41 (3.3) | 70 (3.7) | 0.88 (0.59–1.30) | 0.523 | 0.94 (0.62–1.42) | 0.774 |
| Death | 31 (0.99) | 6 (0.5) | 25 (1.3) | 0.36 (0.15–0.88) | 0.025 | 0.47 (0.19–1.16) | 0.101 |

Continuous variables include median (IQR), and mean. Categorical variables include number (% of total). Odds ratios include (95% CI).

*adjusted for age and sex

**underlying medical condition

[t] maintained significance following Holm-Bonferroni adjustments

between the two groups (OR: 0.96; 95% CI: 0.79–1.17; p-value: 0.687). SARI-positive subjects were more likely to receive antibiotics both before and during hospitalization. No differences were detected between length of stay, ICU admission, oxygen use, mechanical ventilation, or death (Table 1).

Virus detection was more common in SARI-positive patients compared to SARI-negative subjects (OR: 2.84; 95% CI: 2.27–3.56; p-value:<0.001). Including co-detections, SARI-positive

subjects were more likely to have RSV, and HMPV (Table 1). Co-detections were also more prevalent in SARI-positive patients. An analysis of single virus detections only yielded one significant difference with the analysis that included co-detections: SARI-negative patients were more likely to have a single virus infection of HRV, whereas there was no significant difference between groups when co-detections were included following Holm-Bonferroni adjustment (Table 1). There were no differences seen in Ct values by SARI status. MERS-CoV was not detected in any sample.

## The diagnostic accuracy of the SARI criteria and the prevalence of cough and fever for each virus

Overall, the sensitivity of the SARI criteria including co-detections for detecting virus-positive patients was 44%, with a specificity of 78%, PPV of 89.8%, and NPV of 24% (**Figs 1 and 2**). The difference in sensitivity and specificity between single and co-detections for a particular virus was within 10%, but differences were larger changes for PPV and NPV depending on the prevalence of the virus in the study.

SARI criteria showed the highest sensitivity for single HMPV detections (64.8%). The sensitivity of the SARI criteria did not reach above 60% for any other virus. Specificity of the criteria was greatest for RSV detections including RSV and co-detections (66.9%). Specificity was between 58.2% and 63.2% for all other viruses. PPV was greatest for RSV detections at 53.6%, including co-detections, and lowest for single-flu detections at 2.2%. Conversely, NPV was greatest for single-flu detections at 98.8% and lowest for rhinovirus detection, including co-detections, at 59%.

The prevalence of fever was highest in patients with single-flu detection at 90%; whereas those with RSV, including co-detections, had a fever rate of 52.4%. The cough rate was highest in those with single-hMPV and single-RSV detections at 96.1% and 96% respectively. The cough rate was lowest in those with single-adenovirus detections at 56.2%.

## Modified SARI criteria without fever (SARI-NoF) or cough (SARI-NoC)

Modifying the SARI criteria by removing fever or cough changed the diagnostic accuracy for each virus (**Fig 2**). Sensitivity of the SARI-NoF and SARI-NoC criteria was greater for detecting each virus than the original SARI criteria. SARI-NoF had the greatest gains in sensitivity with an increase of 35·1% in detecting any of the viruses, including a 44.1% increase in RSV detection. Conversely, SARI-NoC had a 13.2% increase in sensitivity for detecting any of the viruses and a 2.7% increase for RSV detection.

Specificity for detecting all viruses was decreased for both SARI-NoF and SARI-NoC criteria compared to the original criteria. The specificity of the SARI-NoF criteria for detecting any of the viruses was decreased by 17.8%. The largest decrease was for adenovirus with a change of -32.4%. The specificity for the SARI-NoC criteria decreased by 43.3% for detecting any of the viruses.

The changes in PPV were more modest for the modified criteria. The PPV of the SARI-NoF criteria for any of the viruses decreased by only 0.1%. The PPV for the SARI-NoC criteria for any of the viruses decreased by 10.4%. The NPV of the SARI-NoF criteria increased by 15.5% for any of the viruses; whereas, the SARI-NoC criteria saw a decrease of 8.5% in the same category.

## Stratification of age groups for the SARI criteria

The proportion of virus-positive patients meeting SARI criteria by their age yielded a clear trend, with the youngest patients being less likely to meet criteria (**Figs 1 and 3A**). Data were

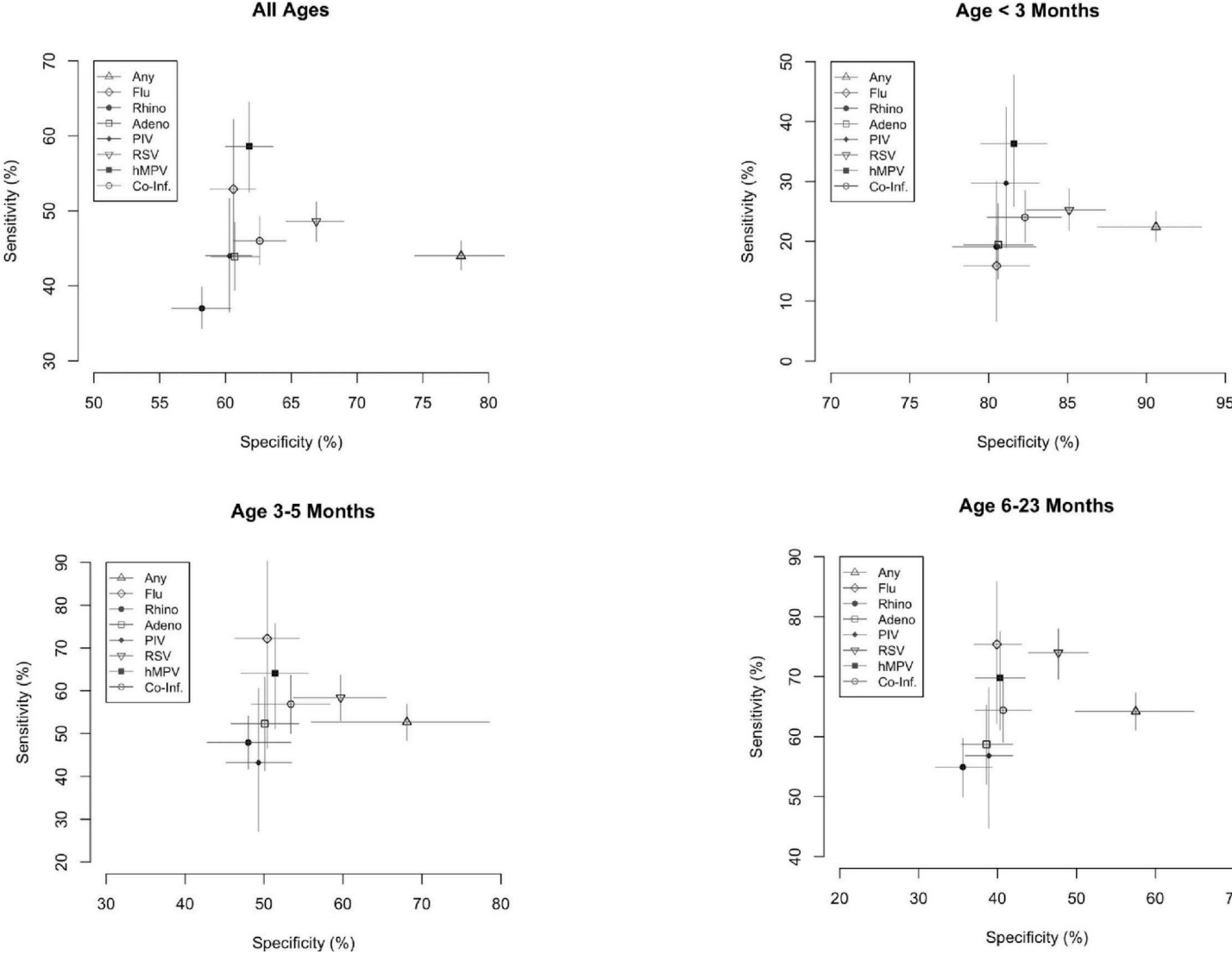

**Fig 1. Sensitivity and specificity of the SARI criteria for each virus.**

stratified into three age groups (<3 months, 3–5 months, 6–23 months) to compare the SARI criteria to the modified criteria in virus-positive patients (**Fig 3A**), showing that the SARI-NoF criteria had the highest sensitivity in all age groups. Differences were also observed with regards to the presence of fever in virus-positive patients across the groups. Virus-positive patients had a reported or measured fever in 44.4%, 60.6%, and 75.5% in the less than 3 months, 3–5 months, 6–23 months groups, respectively (**Fig 3B**). Fever was more common in the virus-negative patients for each age group (**Table 2**).

## The diagnostic accuracy of SARI, SARI-NoF, SARI-NoC criteria for each virus stratified by age

The sensitivity, specificity, PPV, and NPV of both the original and modified criteria for each virus was also stratified by age group (**Figs 4–6**). The SARI criteria had the lowest sensitivity in the youngest age group at 22.4% for detecting any of the viruses. Its specificity in this category,

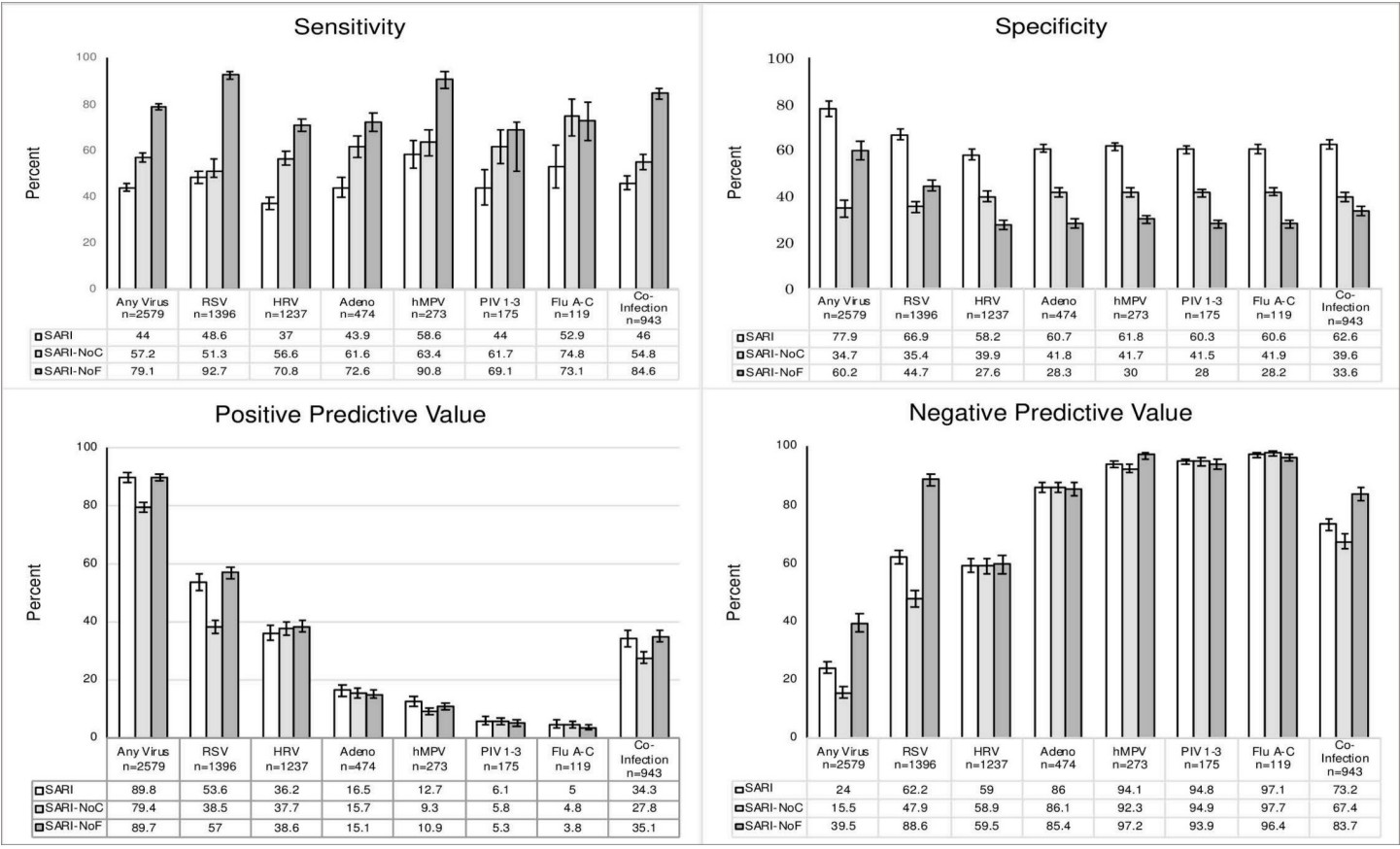

**Fig 2. Sensitivity, specificity, PPV, NPV of original and modified criteria with 95% confidence intervals.**

however, was 90.6%. The SARI criteria had the highest sensitivity in the oldest age group at 64.3% for detecting any of the viruses. The specificity in this age category was 57.7%. The SARI-NoF criteria saw the greatest gains in sensitivity amongst the youngest age groups compared to the original criteria with an increase of 46.6% for detecting virus-positive patients (**Figs 4–6**).

## Discussion

Our study found that for ARI surveillance, the SARI criteria fails to capture more than 50% of children with respiratory virus infections (RVI), and it performs even more poorly for children <3 months. Therefore, it is important to be cautious when using the SARI criteria for surveillance studies, especially if trying to determine the burden of illness to select viruses to influence policy decisions (e.g. vaccine implementation and/or effectiveness). If understanding the true burden of disease is desired, a higher sensitivity is important. While the SARI case definition was initially intended for flu surveillance, several studies have since used it for surveillance of other respiratory viruses [16–18, 22–24]. Our study provides researchers and public health officials the data on diagnostic accuracy of the criteria for detecting ten different respiratory viruses and would caution SARI use for true RVI burden in children.

During our study period, the SARI criteria correctly identified a little over half of the subjects who were flu-positive. Comparisons to previous studies are nuanced due to differing age stratifications, but this result is similar to the 52% sensitivity reported by the Amini (2017)

## a. Meeting SARI and Modified Critiera

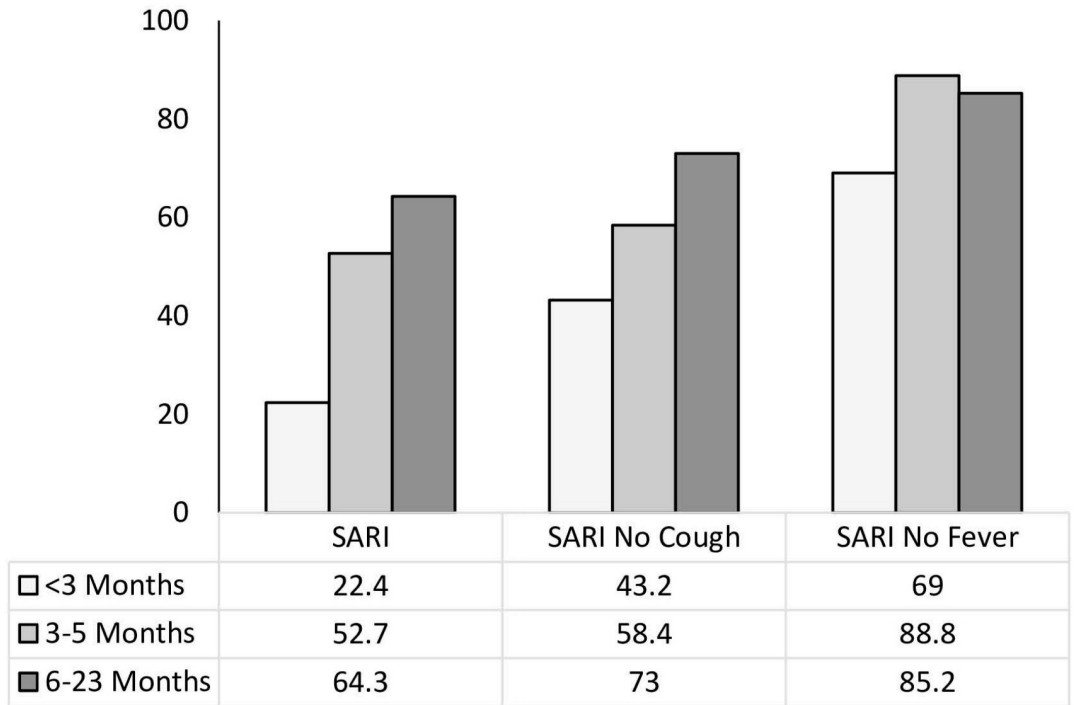

| | SARI | SARI No Cough | SARI No Fever |
|---|---|---|---|
| ☐ <3 Months | 22.4 | 43.2 | 69 |
| ☐ 3-5 Months | 52.7 | 58.4 | 88.8 |
| ☐ 6-23 Months | 64.3 | 73 | 85.2 |

## b. With SARI Symptoms

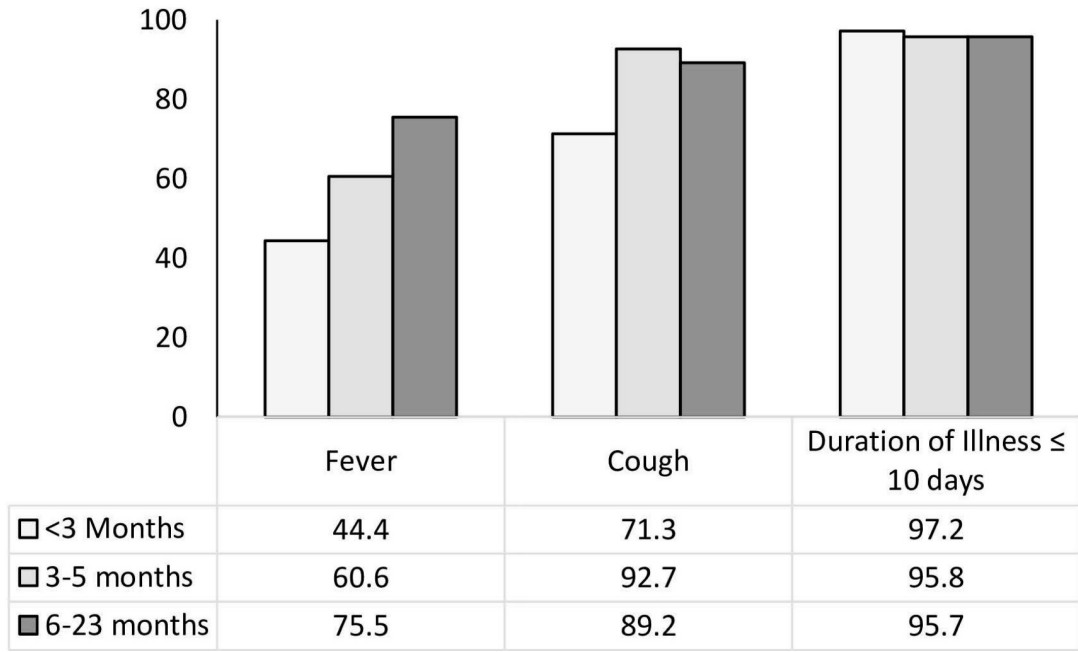

| | Fever | Cough | Duration of Illness ≤ 10 days |
|---|---|---|---|
| ☐ <3 Months | 44.4 | 71.3 | 97.2 |
| ☐ 3-5 months | 60.6 | 92.7 | 95.8 |
| ☐ 6-23 months | 75.5 | 89.2 | 95.7 |

**Fig 3. Virus positive patients stratified by age.**

study in Quebec for children <1 year; however, it is lower than the 79.2% found in that same study for its one to four-year-old age group [14]. Similarly, two studies in Kenya that reported

**Table 2. Characteristics of virus-positive vs. virus-negative patients stratified by age.**

| | <3 Months (n = 1436) | | | | 3–5 Months (n = 617) | | | | 6–23 Months (n = 1111) | | | |
|---|---|---|---|---|---|---|---|---|---|---|---|---|
| | Virus Positive n = 1097 | Virus Negative n = 339 | Adjusted Odds Ratio* | P-Value | Virus Positive n = 545 | Virus Negative n = 72 | Adjusted Odds Ratio* | P-Value | Virus Positive n = 937 | Virus Negative n = 174 | Adjusted Odds Ratio* | P-Value |
| Meets SARI Criteria | 245 (22.3) | 32 (9.44) | 2.78 (1.88–4.12) | <0.001[t] | 286 (52.5) | 23 (31.9) | 2.37 (1.41–4.01) | 0.001[t] | 601 (64.1) | 74 (42.5) | 2.42 (1.74–3.36) | <0.001[t] |
| Meets SARI-NoC Criteria | 474 (43.2) | 211 (62.2) | 0.465 (0.362–0.597) | <0.001[t] | 318 (58.4) | 43 (59.7) | 0.959 (0.58–1.58) | 0.869 | 684 (73.0) | 128 (73.6) | 0.975 (0.676–1.41) | 0.893 |
| Meets SARI-NoF Criteria | 757 (69.0) | 83 (24.5) | 6.85 (5.18–9.05) | <0.001[t] | 484 (88.8) | 47 (65.3) | 4.19 (2.41–7.29) | <0.001[t] | 798 (85.2) | 103 (59.2) | 3.96 (2.78–5.64) | <0.001[t] |
| Fever reported | 428 (39.0) | 207 (61.1) | 0.410 (0.319–0.526) | <0.001[t] | 293 (53.8) | 40 (55.6) | 0.938 (0.57–1.54) | 0.801 | 667 (71.2) | 127 (73.0) | 0.916 (0.637–1.32) | 0.636 |
| Fever > = 38C | 366/1095 (33.4) | 169 (49.9) | 0.509 (0.397–0.651) | <0.001[t] | 223/544 (41.0) | 32/71 (45.1) | 0.860 (0.52–1.42) | 0.554 | 492/934 (52.7) | 108/173 (62.4) | 0.671 (0.48–0.94) | 0.019 |
| Any Fever | 487 (44.4) | 213 (62.8) | 0.476 (0.37–0.61) | <0.001[t] | 330 (60.6) | 43 (59.7) | 1.05 (0.64–1.73) | 0.852 | 707 (75.5) | 136 (78.2) | 0·862 (0.58–1.27) | 0.454 |
| Cough | 782 (71.3) | 88 (26.0) | 7.06 (5.36–9.30) | <0.001[t] | 505 (92.7) | 50 (69.4) | 5.51 (3.03–10.0) | <0.001[t] | 836 (89.2) | 110 (63.2) | 4·84 (3.33–7.02) | <0.001[t] |
| Duration of Symptoms | 3 (2–4) 3.38 | 2 (1–3) 2.72 | 1·10 (1.04–1.16) | 0.001[t] | 3 (2–5) 4.32 | 3 (2–4) 3.71 | 1.05 (0.97–1.15) | 0.212 | 3 (2–5) 5.09 | 3 (1–5) 4.0 | 1.01 (0.98–1.05) | 0.493 |

Continuous variables include median (IQR), and mean.

Categorical variables include number (% of total). Odds ratios include (95% CI)

*adjusted for sex

[t]maintained significance following Holm-Bonferroni correction

results for participants aged two months to four years found sensitivities of 84.3% and 89.6% [15, 24]. The specificity of the criteria for detecting flu in our study (60.6%) also differs from what was found in the aforementioned studies (13.1%-29.5%). The age distribution differences likely explain these discrepancies as our cohort consists only of children under two years, and the SARI criteria were the least sensitive (15.9%) and most specific (80.5%) for detecting flu in our patients under three months, who make up 45.4% of our study population. The sensitivities and specificities of the criteria for our patients aged 3–5 months (72.2%, 50.4%) and 6–23 months (75.4%, 39.9%) are more closely aligned with the findings of the previous studies. Therefore, if policy makers are deciding if flu vaccine should be administered to pregnant women or infants based on the SARI definition, they would be underestimating the true burden of flu illness.

Our findings for the diagnostic accuracy of the criteria for detecting RSV also have some differences from previous studies, including one study that included similar age group stratifications—allowing for a more direct comparison [17]. For children <3 months, our study showed a sensitivity of 25.2%, lower than the 55% found by Rha (2018) in South Africa [17]. However, our 74% sensitivity in the 6–23 months age group more closely matched their groups of 6–11 and 12–23 months with sensitivities of 77% and 81%, respectively. The latter result is also in line with the findings of Nyawanda (2016), which reported sensitivities for children <1 year at 79.4%, and one to four years at 86.2%. Our finding for the specificity of the criteria was greater than these two previous studies across all age groups but differed most from Rha for children less than three months (85% vs 54%). Similar to Rha, however, the specificity of the criteria in our study for our oldest age group (6–23 months, 47.7%) was nearly half that of our

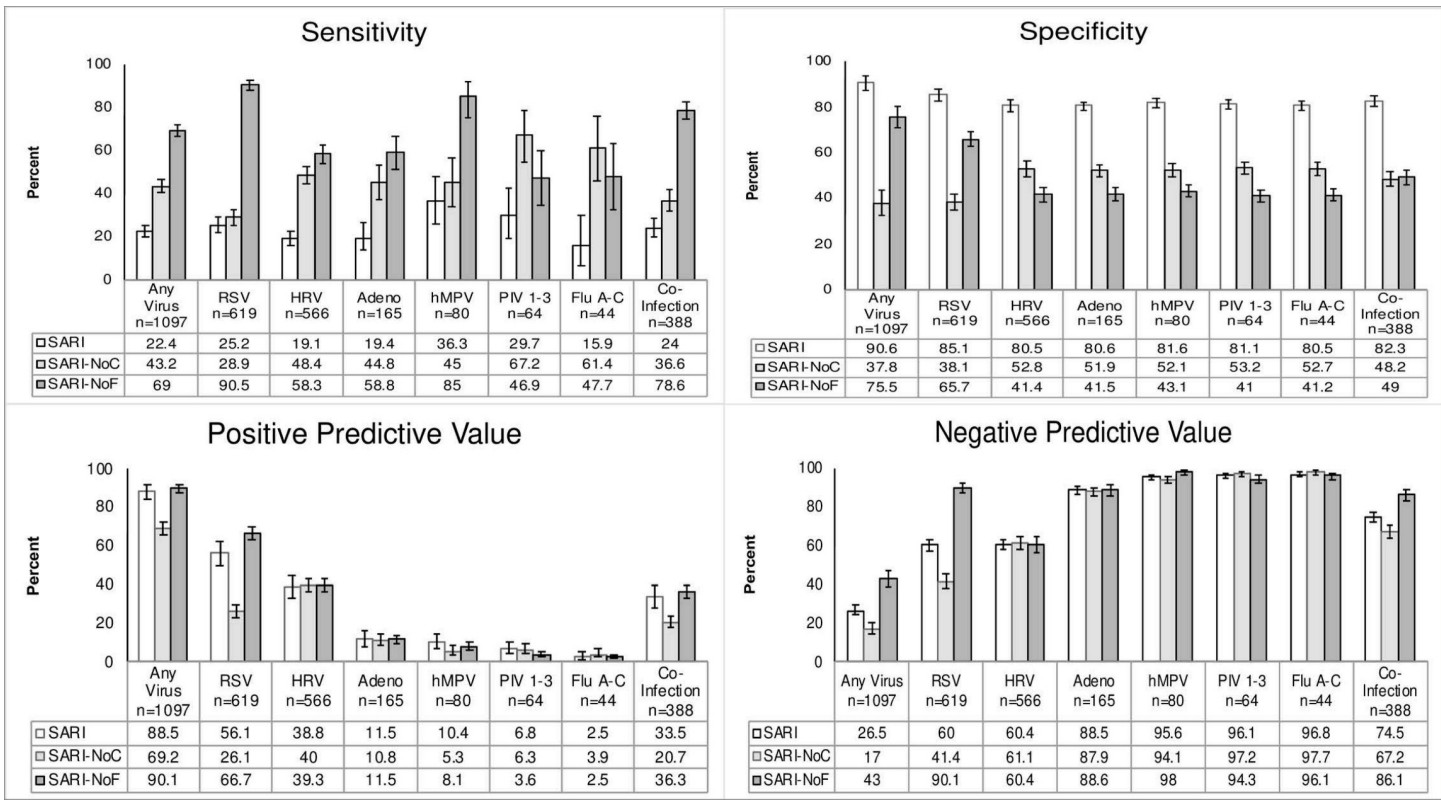

**Fig 4. Sensitivity, specificity, PPV, NPV of original and modified criteria with 95% confidence intervals in subjects less than 3 months old.**

youngest age group (85.1%). One potential explanation for differences between our studies is the use of singleplex RT-PCR in our study versus the multiplex RT-PCR that was used in Rha, with singleplex historically being more sensitive. Given that RSV vaccines are in the pipeline, it is important to also capture all cases of RSV to determine the true burden of illness to document if the vaccine is effective at impacting disease. Thus, if the SARI definition is used for ARI surveillance studies, it will once again underestimate the burden of RSV illness, and therefore altering the inclusion criteria to include either fever and/or cough may be more accurate.

The differences in diagnostic accuracy of the SARI criteria by age is the most significant finding of this study. The SARI criteria performed poorly with regards to sensitivity for the detection of any respiratory virus in children under three months. Only 22.4% of children in this age group with a respiratory virus were captured by the case definition. This increased to 64.3% in children aged 6–23 months. Conversely, the criteria are very specific in the youngest age group for detecting any respiratory virus compared to the oldest age group (90.6% vs. 64.3%).

We further dissected by symptoms and found that having the combination of both fever and cough was the reason for low sensitivities and having either one or the other could capture more RVI cases. Specifically, the lower rates of fever—and to a lesser extent, cough—in patients <3 months who are respiratory virus-positive account for the poor sensitivity of the SARI criteria in this age group. Only 44.4% of the virus-positive patients <3 months had a reported or measured fever, compared to 75.5% of their 6–23 months counterparts. Consistent with what has been found in previous studies [25], removing fever from the SARI criteria greatly increased the sensitivity of detecting any respiratory virus in children <3 months,

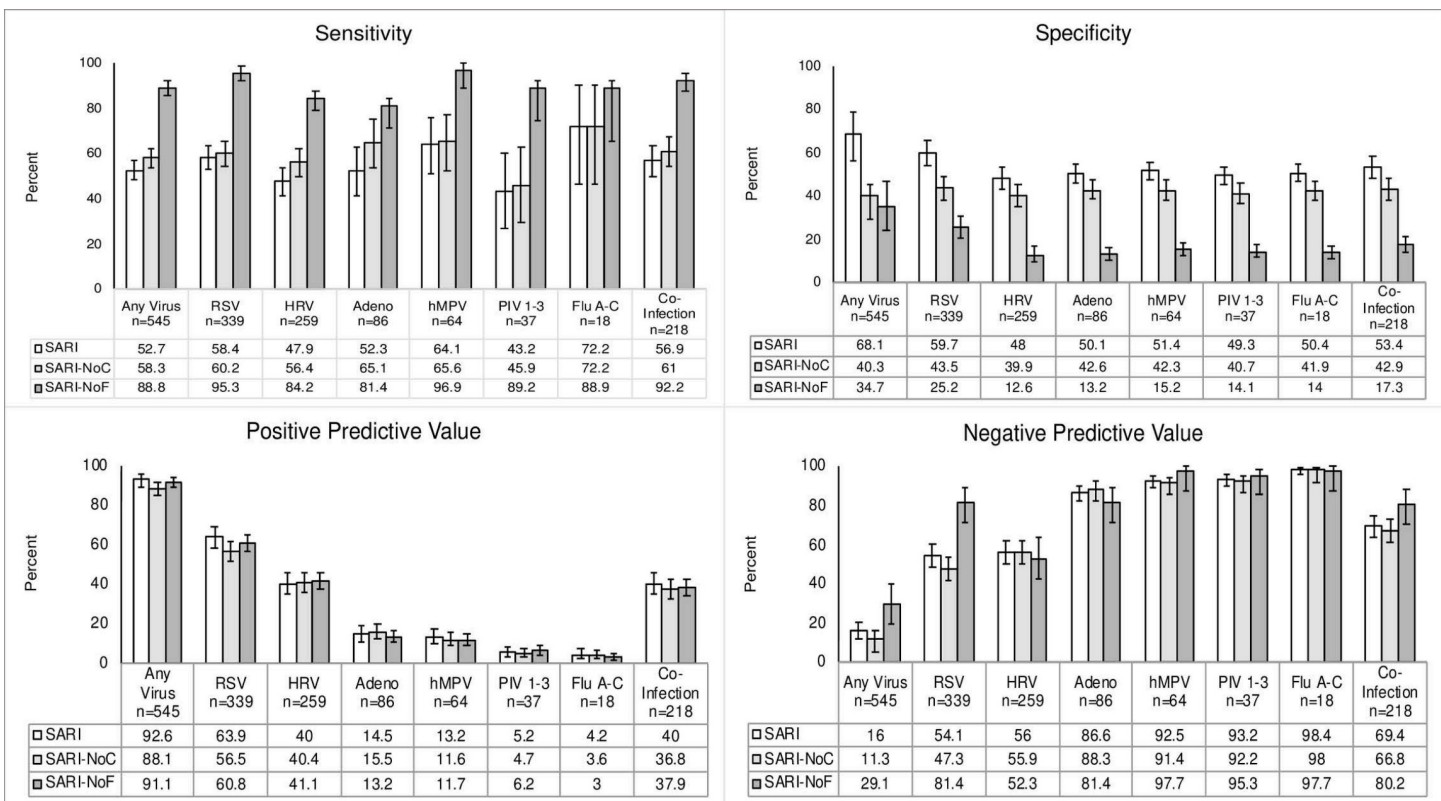

**Fig 5. Sensitivity, specificity, PPV, NPV of original and modified criteria with 95% confidence intervals in subjects 3–5 months old.**

particularly for RSV, which then increased by 65.3% [18]. Notably, both the PPV and NPV for detecting RSV also increased with this change. In general, for this youngest age group, removing fever as a criterion increases sensitivity more than it decreases specificity for each virus tested, and it results in minimal changes to PPV and NPV. This result would seem to advise caution when mandating that fever must be included as a criterion for this age group.

Our study has several strengths. It is the first to report the diagnostic accuracy of the WHO SARI criteria for detecting multiple viruses, including but not limited to flu and RSV. It is also the first study of this kind in the Middle East. Our prospective study took place over three years, included very young children, and included both fever and/or respiratory criteria, which allowed us to compare SARI-positive to SARI-negative children. Our study also has a number of limitations. It only includes hospitalized children <2 years who had ARI symptoms from one large hospital in Amman, Jordan. This limits its generalizability to other locations, or to study populations that include all hospitalized patients. Additionally, the 2011 SARI criteria were developed during our study period. While our study design allowed us to capture all the necessary elements to determine which patients met SARI criteria, it involved combining together the data as opposed to a straightforward data collection tool. Part of the data used in this study also included a parental survey of symptoms—introducing a source of reporting bias.

## Conclusions

The SARI criteria were initially developed by the WHO in 2011 as part of its recommendations for global flu surveillance. The ultimate goal of such surveillance is "to minimize the impact of

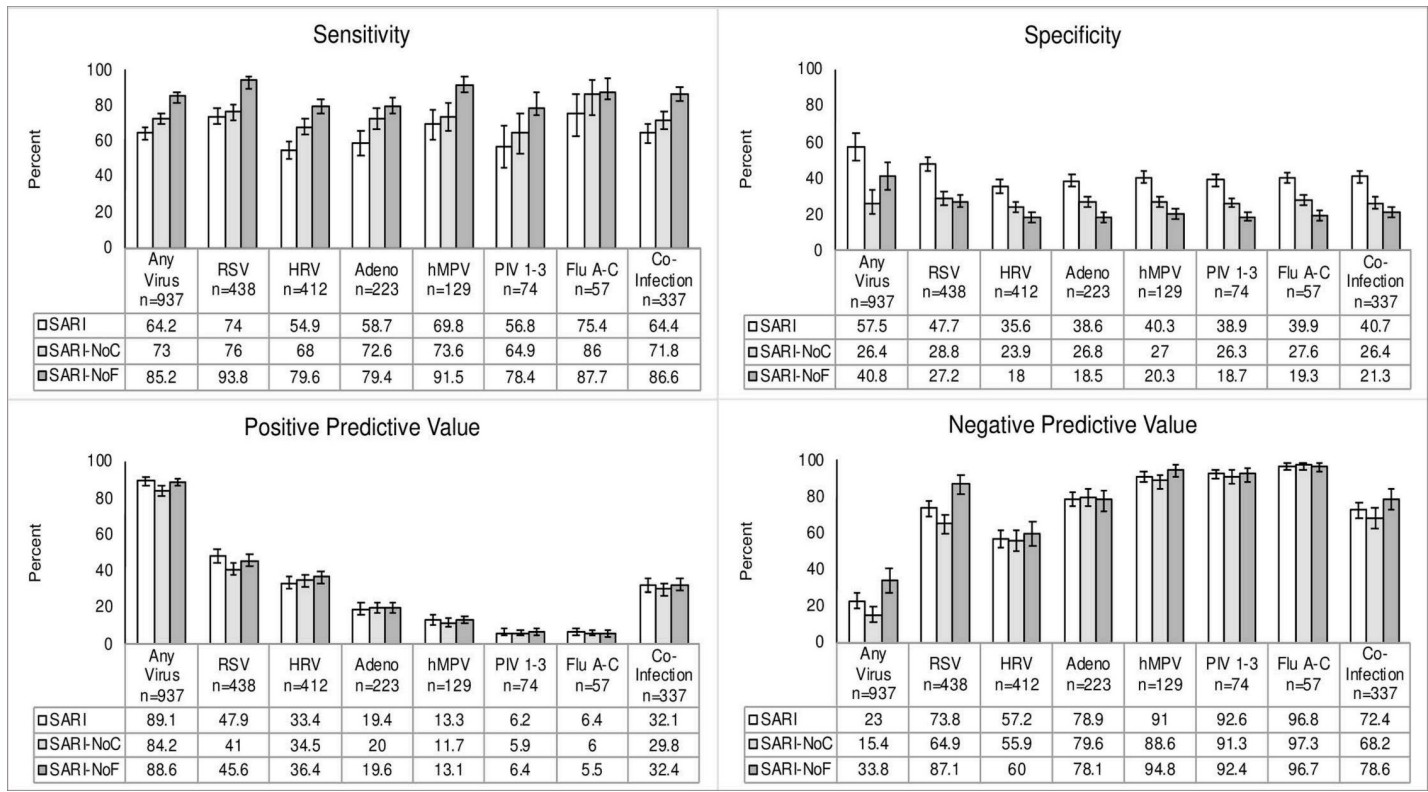

**Fig 6. Sensitivity, specificity, PPV, NPV of original and modified criteria with 95% confidence intervals in subjects 6–23 months old.**

the disease by providing useful information to public health authorities so they may better plan appropriate control and intervention measures, allocate health resources, and make case management recommendations" [4]. However, our study found that more than half of the RVI cases would have been missed using the SARI criteria. Therefore, it is critical to know how well these criteria perform in different patient populations and geographic settings. Our study shows that use of the criteria in children <3 months significantly underestimates the burden of disease for RSV, flu, and other viruses. Therefore, removing fever as a mandatory criterion in this age group, particularly for RSV, would greatly increase sensitivity with an acceptable decrease in specificity. Overall, more studies need to be conducted in the Middle East to gather more information on this finding; however, it is advisable that policymakers are cautious when using the SARI criteria.

## Supporting information

**S1 Dataset.**
(CSV)

## Author Contributions

**Conceptualization:** Thomas Klink.

**Data curation:** Danielle A. Rankin, Bhinnata Piya.

**Formal analysis:** Thomas Klink, Andrew J. Spieker.

**Investigation:** Samir Faouri, Asem Shehabi.

**Methodology:** John V. Williams, Najwa Khuri-Bulos, Natasha B. Halasa.

**Project administration:** Samir Faouri, Natasha B. Halasa.

**Supervision:** Andrew J. Spieker, Najwa Khuri-Bulos, Natasha B. Halasa.

**Writing – original draft:** Thomas Klink, Natasha B. Halasa.

**Writing – review & editing:** Thomas Klink, Danielle A. Rankin, Bhinnata Piya, Andrew J. Spieker, Samir Faouri, Asem Shehabi, John V. Williams, Najwa Khuri-Bulos, Natasha B. Halasa.

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
