## [Decision Letter · Decision Letter 0]

23 Jan 2020

PONE-D-19-32487

Evaluating the Diagnostic Accuracy of the WHO Severe Acute Respiratory Infection (SARI) Criteria in a Middle Eastern Pediatric Cohort over Three Respiratory Seasons

PLOS ONE

Dear Dr. Halasa,

Thank you for submitting your manuscript to PLOS ONE. After careful consideration, we feel that it has merit but does not fully meet PLOS ONE’s publication criteria as it currently stands. Therefore, we invite you to submit a revised version of the manuscript that addresses the points raised during the review process.

The Authors are expected to address all the criticisms by all Reviewers. In particular, please cite the relevant odds ratios when describing the results, tone down the conclusion concerning advice to policy maker and researchers (Reviewer #1), and assess if the calculation of NPV was based on true negatives (Reviewer #2). In additional to the above comments, please address,

Title: to reflect the target study population, please add something like ‘for children aged under 2 years’Table 1, please clarify the term ‘premature’Figure 6, to avoid confusion, please use ‘SARI-NoF’ and ‘SARI-NoC’ for ‘No Fever’ and ‘No Cough’

We would appreciate receiving your revised manuscript by Mar 08 2020 11:59PM. To enhance the reproducibility of your results, we recommend that if applicable you deposit your laboratory protocols in protocols.io, where a protocol can be assigned its own identifier (DOI) such that it can be cited independently in the future. For instructions see: http://journals.plos.org/plosone/s/submission-guidelines#loc-laboratory-protocols

We look forward to receiving your revised manuscript.

Kind regards,

Eric HY Lau, Ph.D.

Academic Editor

PLOS ONE

Additional Editor Comments (if provided):

The Authors are expected to address all the criticisms by all Reviewers. In particular, please cite the relevant odds ratios when describing the results, tone down the conclusion concerning advice to policy maker and researchers (Reviewer #1), and assess if the calculation of NPV was based on true negatives (Reviewer #2). In additional to the above comments, please address,

1. Title: to reflect the target study population, please add something like ‘for children aged under 2 years’

2. Table 1, please clarify the term ‘premature’

3. Figure 6, to avoid confusion, please use ‘SARI-NoF’ and ‘SARI-NoC’ for ‘No Fever’ and ‘No Cough’

Journal Requirements:

Reviewers' comments:

Reviewer's Responses to Questions

**Comments to the Author**

1. Is the manuscript technically sound, and do the data support the conclusions?

Reviewer #1: Yes

Reviewer #2: Yes

2. Has the statistical analysis been performed appropriately and rigorously? 

Reviewer #1: Yes

Reviewer #2: Yes

3. Have the authors made all data underlying the findings in their manuscript fully available?

Reviewer #1: No

Reviewer #2: No

4. Is the manuscript presented in an intelligible fashion and written in standard English?

Reviewer #1: Yes

Reviewer #2: Yes

5. Review Comments to the Author

Reviewer #1: This is an important manuscript and needed for filling the literature gap in Middle East.

I have a few comments concerning presentation of your methods and findings.

Page 4, sentence 65, you mention that inclusion and exclusion criteria are mentioned elsewhere, as a reader I would like to see them to understand the context of your research.

Page 4, sentence 77-78, you mention a standardized questionnaire however nothing more about the variables used and how different they are from the case-reporting form is mentioned. It would be worth documenting what the questionnaire included.

In same sentences, you mention the personnel filling the questionnaires did so by asking in Arabic but recording in English, were the terms from Arabic to English standardized for limiting errors?

Page 8, table 1, it is not clear that Age here is presented as Median, tables should be understood on their own, it would be worth adding (median) next to the variable Age.

Page 9, sentences 133-149, it is not clear that ORs are being explained in this text, it would be worth putting ORs in parenthesis or reminding readers to refer to the tables for more information. It is just mentioned once at the start of the sentence 132.

In your conclusion, page 16, you state: "we advise caution for policy-makers and researchers when using the SARI criteria as currently written to inform the utilization of prevention measures like vaccines", however this contradicts with your discussion limits on the fact that this study was done in one hospital in Jordan and hence can not be generalized. It is advisable to state that more studies need to be done in the Middle East for more information on this finding however it is advisable that policy makers are cautious when using SARI criteria.

Reviewer #2: The paper is written well and has findings that are important for studies on respiratory diseases in children. All the findings are shown in odd ratios I suggest that in some of the results are more intuitive if they were proportions. The odds of fulfilling SARI case definition among those that are submitted with respiratory symptoms or fever is more difficult to understand. It would also be good to give more understanding on what the other cases are – e.g. would a lot be asthmatic? The proportion of any virus positive is very high – with more than 80% positivity, were there no bacterial infections and asthma cases among those that had either respiratory symptoms or fever? The way the virus detection is presented in the SARI positive ones is not really helpful. Further for the analysis of the negative predictive value in the table the number of influenza positives is mentioned, but that calculation should be based on the true negative cases.

6. PLOS authors have the option to publish the peer review history of their article (what does this mean?). If published, this will include your full peer review and any attached files.

Reviewer #1: No

Reviewer #2: No

---

## [Author Response · Author response to Decision Letter 0]

10 Mar 2020

March 8, 2020

Plos One: Revision [PONE-D-19-32487] “Evaluating the Diagnositc Accuracy of the WHO Severe Acute Respiratory Infection (SARI) Criteria in a Middle Eastern Pediatric Cohort over Three Respiratory Seasons.

Dear Editors:

We appreciate this opportunity to respond to the comments of the editor and the reviewers for our manuscript. We have outlined each of the modifications made in the accordance with these comments. The comments of the reviewers are in bold and the responses are below each comment. 

The Authors are expected to address all the criticisms by all Reviewers. In particular, please cite the relevant odds ratios when describing the results, tone down the conclusion concerning advice to policy maker and researchers (Reviewer #1), and assess if the calculation of NPV was based on true negatives (Reviewer #2). In additional to the above comments, please address,

1. Title: to reflect the target study population, please add something like ‘for children aged under 2 years’

RESPONSE:

We have changed the title to “Evaluating the Diagnostic Accuracy of the WHO Severe Acute Respiratory Infection (SARI) Criteria in Middle Eastern Children Under Two Years over Three Respiratory Seasons”

2. Table 1, please clarify the term ‘premature’

RESPONSE:

We have added <37 weeks to the table. 

3. Figure 6, to avoid confusion, please use ‘SARI-NoF’ and ‘SARI-NoC’ for ‘No Fever’ and ‘No Cough’

RESPONSE:

We have made these changes. 

Reviewer #1: This is an important manuscript and needed for filling the literature gap in Middle East. I have a few comments concerning presentation of your methods and findings. Page 4, sentence 65, you mention that inclusion and exclusion criteria are mentioned elsewhere, as a reader I would like to see them to understand the context of your research.

RESPONSE:

Thank you for your comment.

We added the following clarification “one of the following admission diagnoses: ARI, apnea, asthma exacerbation, bronchiolitis, bronchopneumonia, croup, cystic fibrosis exacerbation, febrile seizure, fever without localizing signs, respiratory distress, pneumonia, pneumonitis, pertussis, pertussis-like cough, rule out sepsis, upper respiratory infection (URI), or other. Children were excluded only if they had chemotherapy-associated neutropenia and/or were newborns who had never been discharged.”

Page 4, sentence 77-78, you mention a standardized questionnaire however nothing more about the variables used and how different they are from the case-reporting form is mentioned. It would be worth documenting what the questionnaire included.

REPSONSE:

We have revised the sentence to this: “Demographic, social, and medical histories were obtained by standardized questionnaires, which were a component of the interview portion of the case report form. All interviews were conducted in Arabic to parents and recorded in English.”

In same sentences, you mention the personnel filling the questionnaires did so by asking in Arabic but recording in English, were the terms from Arabic to English standardized for limiting errors?

REPSONSE:

Yes, we included Arabic and English in our Case Report Forms to prevent errors. 

Page 8, table 1, it is not clear that Age here is presented as Median, tables should be understood on their own, it would be worth adding (median) next to the variable Age.

RESPONSE:

Added.

Page 9, sentences 133-149, it is not clear that ORs are being explained in this text, it would be worth putting ORs in parenthesis or reminding readers to refer to the tables for more information. It is just mentioned once at the start of the sentence 132.

REPONSE:

This has been done.

In your conclusion, page 16, you state: "we advise caution for policy-makers and researchers when using the SARI criteria as currently written to inform the utilization of prevention measures like vaccines", however this contradicts with your discussion limits on the fact that this study was done in one hospital in Jordan and hence can not be generalized. It is advisable to state that more studies need to be done in the Middle East for more information on this finding however it is advisable that policy makers are cautious when using SARI criteria.

REPSONSE:

We have changed the statement as suggested to “Overall, more studies need to be conducted in the Middle East to gather more information on this finding; however, it is advisable that policy-makers are cautious when using the SARI criteria.”

Reviewer #2: The paper is written well and has findings that are important for studies on respiratory diseases in children. All the findings are shown in odd ratios I suggest that in some of the results are more intuitive if they were proportions. 

REPSONSE:

We need further clarification about which specific results would make more sense as proportions than odds ratios. 

The odds of fulfilling SARI case definition among those that are submitted with respiratory symptoms or fever is more difficult to understand. It would also be good to give more understanding on what the other cases are – e.g. would a lot be asthmatic? 

REPSONSE:

Only 4.61% had asthma exacerbation/reactive airway diagnoses as an admission diagnoses, with less than three percent of all children having the diagnoses. Of those, only 45% met SARI criteria. If the reviewer wants us to add it to the table, we are happy to do so.

The proportion of any virus positive is very high – with more than 80% positivity, were there no bacterial infections and asthma cases among those that had either respiratory symptoms or fever? 

Only 4.61% had asthma exacerbation/reactive airway diagnoses as an admission diagnoses, with less than three percent of all children having the diagnoses. Of those, only 45% met SARI criteria.

Reports of blood, urine and CSF cultures were available for only 764, 769 and 614 subjects, respectively. Of these subjects, 38/764 (5.0%) had positive blood cultures, 118/769 (15.3%) had positive urine cultures and 4/614 (0.7%) had positive CSF cultures. Of the 150 subjects who tested positive for at least one bacterial culture, 104 (69.3%) also had viral codetection. Therefore, we do not believe it will be useful to look at SARI criteria in these subjects due to low sample sizes. 

REPSONSE:

The way the virus detection is presented in the SARI positive ones is not really helpful. 

REPSONSE:

Further clarification of which figure and/or table the reviewer is referring to would be helpful.

Further for the analysis of the negative predictive value in the table the number of influenza positives is mentioned, but that calculation should be based on the true negative cases.

REPSONSE:

We calculated the negative predictive value by the true negatives divided by total number of people who tested negative. We are a bit confused on this comment and would like some additional clarification to address the reviewer’s comment.

Sincerely,

Natasha Halasa, MD MPH

Associate Professor, Pediatric Infectious Diseases

Vanderbilt University Medical Center

---

## [Editor Report · Decision Letter 1]

9 Apr 2020

Evaluating the diagnostic accuracy of the WHO severe acute respiratory infection (SARI) criteria in middle eastern children under two years over three respiratory seasons

PONE-D-19-32487R1

Dear Dr. Halasa,

We are pleased to inform you that your manuscript has been judged scientifically suitable for publication and will be formally accepted for publication once it complies with all outstanding technical requirements.

With kind regards,

Eric HY Lau, Ph.D.

Academic Editor

PLOS ONE
---

## [Editor Report · Acceptance letter]

14 Apr 2020

PONE-D-19-32487R1 

Evaluating the diagnostic accuracy of the WHO severe acute respiratory infection (SARI) criteria in middle eastern children under two years over three respiratory seasons 

Dear Dr. Halasa:

I am pleased to inform you that your manuscript has been deemed suitable for publication in PLOS ONE. Congratulations! Your manuscript is now with our production department. 

With kind regards,

on behalf of

Dr. Eric HY Lau 

Academic Editor

PLOS ONE